# It's not black and white: Perspectives of Western Canadian beef farmers on dairy-beef production

Bianca Vandresen📵, Daniel M. Weary, Marina A. G. von Keyserlingk📵*

Animal Welfare Program, Faculty of Land and Food Systems, The University of British Columbia, Vancouver, British Columbia, Canada

* nina@mail.ubc.ca

## Abstract

Non-replacement dairy calves (i.e., males and females not needed for milking herd replacement) can face multiple welfare challenges due to their low economic value in the dairy and beef industries. Incorporating beef genetics into dairy herd breeding programs has become common to produce beef-on-dairy crossbred calves that are better suited for beef production than pure dairy breed animals. This practice has the potential to increase revenue from non-replacement dairy calves for dairy farmers, but little is known about its impact on beef farmers. This study aimed to investigate the attitudes of Canadian beef producers toward dairy-beef production, with a focus on how beef-on-dairy breeding strategies by dairy may affect the beef industry. We conducted semi-structured interviews with 20 beef farmers in Western Canada, exploring their awareness, attitudes, and recommendations for the management of beef-on-dairy calves. Participants (11 male, 9 female) were recruited using snowball sampling and interviewed following a semi-structured interview guide. The audio-recorded interviews (averaging $44 \pm 15$ min in duration) were transcribed verbatim and analyzed using inductive thematic analysis, resulting in three main themes: 1) the dairy and beef relationship, 2) attitudes to beef-on-dairy animals, and 3) a shared future. In the first theme, participants discussed the relationship between the dairy and beef industries, highlighting differences in Canadian market structures (dairy as supply-managed *vs.* beef as an open market), farming practices (beef as more extensive *vs.* dairy as more intensive) and public perceptions of the two systems. In the second theme, participants showed mixed attitudes toward dairy-beef production and discussed their views about beef-on-dairy calves compared to purebred dairy calves, the management practices used to raise them, and the potential impacts of dairy-beef production on the beef industry. In the third theme, participants reflected on the future of dairy-beef production, discussing who should be involved in shaping the future of this practice. Participants showed mixed feelings towards the use of beef genetics in dairy herds, with some perceiving this as an opportunity for the beef industry to meet consumer demand and others expressing concern about the over-saturation

---

**Data availability statement:** https://doi.org/10.5683/SP3/SD4SSJ.

**Funding:** Social Sciences and Humanities Research Council of Canada The funders had no role in study design, data collection and analysis, decision to publish, or preparation of the manuscript.

**Competing interests:** No authors have competing interests.

of the beef market and possible threats to traditional ways of rearing beef. Our study enhances the understanding of the relationship between the dairy and beef industries in Western Canada and suggests the need for communication and collaboration among producers and others in the supply chain.

## Introduction

The dairy industry's primary saleable product is milk, and dairy cows have been selected for high milk yields. Cows must give birth to produce milk (typically around once a year), but not all calves are needed as replacements in dairy herds [1]; almost all male calves and any excess females can be termed 'non-replacements' (or 'surplus' or 'excess' calves; see [2]). Globally, these calves are managed through one of three pathways: early life killing, raised for veal, or raised for beef; see [3]. In Canada, many non-replacement dairy calves are sold in the first few weeks of life to veal producers, a pathway that includes multiple welfare challenges [4], including often receiving lower amounts of colostrum and milk compared to heifer calves raised as replacements for the milking herd [5–7]. In some cases, dairy farmers may euthanize non-replacement calves shortly after birth if their economic value does not justify the rearing costs. A 2017 survey of Canadian dairy farmers found that 5% reported euthanizing at least one male calf in the previous year. Both early-life killing and the issues related to the management of non-replacement dairy calves have drawn public scrutiny and media attention (e.g., [8,9]), challenging the social sustainability of the dairy industry.

The use of beef genetics in dairy herds to produce dairy-beef crossbred calves (hereafter referred to as 'beef-on-dairy breeding strategies') has been facilitated by the availability of sexed semen to produce replacement pure-bred female dairy calves from selected cows in the herd, and beef-on-dairy breeding to produce non-replacement calves that are of higher value for the beef market from the remaining cows in the herd [10]. This approach provides an additional source of revenue for dairy farmers and improves the potential of rearing these non-replacement calves for beef [11]. In this study, 'dairy-beef production' refers to all animals raised on dairy farms that enter the beef supply chain, regardless of breed or age (e.g., purebred dairy calves and beef-on-dairy calves, as well as cull dairy cows). The term 'beef-on-dairy' specifically denotes calves produced through beef × dairy crossbreeding, and the term 'non-replacement dairy calves' refers to all calves bred on dairy farms that are not needed as replacements for the milking herd, which can be purebred dairy calves or beef-on-dairy calves.

Public opinions on the management of non-replacement dairy calves have been a key driver of debates regarding alternative practices in dairy farming [8,12], and these perspectives may fall out of step with routine dairy calf management practices [13]. For example, the immediate separation at birth of the calf from its mother (i.e., early cow-calf separation) is common on dairy farms, but widely regarded as unacceptable by the general public [14]. When considering beef-on-dairy production practices,

early cow-calf separation is relevant because non-replacement dairy calves are separated from their dams shortly after birth; whereas, calves reared on cow-calf farms are typically raised with their dams until weaning at approximately 6–8 months of age [15].

The supply of dairy animals into beef production (i.e., dairy-beef production) is not new. For example, 20% of Canadian beef comes from dairy farms, largely related to cull dairy cows being slaughtered for beef [16]. However, the adoption of beef-on-dairy breeding strategies is likely to affect the relationship between the beef and dairy industries. One of the few studies addressing this topic reported that Irish beef farmers were resistant to raising beef-on-dairy calves, citing concerns such as poor profit margins, low-quality calves, and market uncertainties [17]. In Canada, a focus group study with dairy farmers and allied industry partners reported that participants believed that beef farmers have no interest in raising beef-on-dairy calves [18]. Members of post-farm gate organizations along the value chain in Australia shared concerns regarding the need to address the non-replacement dairy calf issue and stressed the importance of collaboration, leadership, and commitment from all sectors contributing to beef production [3].

Beef farmers (also frequently referred to as ranchers) may be impacted by the increased supply of dairy beef due to the adoption of beef-on-dairy breeding strategies. However, to our knowledge, beef farmers' perspectives on beef-on-dairy animals and associated calf-rearing practices remain underexplored. To address this gap, this study aimed to explore the attitudes of Canadian beef farmers toward dairy-beef production, focusing on their perspectives on beef-on-dairy breeding, the future of this practice and how it may impact beef farmers.

## Methods

This study was approved by The University of British Columbia (UBC) Behavioral Research Ethics Board (protocol H23-00351). Interviews were conducted over the phone, and all participants provided verbal consent to participate.

### Sampling and participant recruitment

There is an ongoing debate about the determination of sample size and data saturation in qualitative research. Some authors propose methods to determine data saturation, while others highlight limitations in the concept of data saturation within qualitative research [19,20]. One of the primary challenges is that determining data saturation requires conducting data analysis, making it virtually impossible to establish the sample size in advance of data collection. Consequently, the concept of data saturation can be regarded as a *post hoc* rationale, with the determination of sample sizes in qualitative research relying instead on interpretative, context-dependent, and pragmatic judgments [21]. Researchers can face multiple logistical challenges when recruiting participants in qualitative research [22]. Despite social media platforms being increasingly widespread among farming communities, social networks remain the primary means of communication to reach farmers [23]. Snowball sampling is a common social network-based sampling method used in qualitative research [24] and is suitable for farmer recruitment. Conducting interviews can present challenges, hindering participant recruitment, as participants must be willing to dedicate time to engage with researchers. Phone interviews provide greater flexibility in scheduling, especially when working with populations in rural areas that are often hard to reach for in-person interviews.

Given these limitations, our study used a snowball method to recruit a convenience sample of participants for phone interviews; recruitment ceased once all known referral paths had been exhausted. Initially, a small number of participants were recruited through the authors' existing networks, and additional participants were identified through recommendations from those who had completed interviews. To be eligible, participants had to make income from beef production and reside in Canada. Since local context can influence peoples' views toward animal production systems [25], we recruited only from within Canada to reflect the culture and market dynamics specific to this region. Recruitment began on September 1, 2023, and ended on May 10, 2024. Despite the efforts to include farmers from across Canada, including offering interviews in French to accommodate participants from francophone regions, the final sample consisted of participants

from three western provinces: Alberta, British Columbia, and Saskatchewan, which are the provinces with the highest concentration of beef production in Canada (Fig 1) [26]. All participants were offered a $10 digital gift card as thanks for participation; one participant declined the gift card.

## Semi-structured interview guide

Semi-structured interviews use pre-determined questions aimed at investigating participants' views about the research topic while allowing flexibility for new questions to be asked depending on what the interviewee shares [27]. This method of research was chosen because it allows for the understanding of underexplored research topics, posing few restraints on the topics covered during the interviews. We developed the semi-structured interview guide to address the research questions and facilitate consistent conversations across interviews, but the specific questions were flexible to promote a comfortable and engaging dialogue. Questions were first developed to address all dairy animals supplied to beef production (i.e., cull dairy cows, veal calves, and dairy-beef calves), based on the reasoning that the beef farmer participants would have low awareness of the use of beef-on-dairy breeding strategies and feel uncomfortable discussing this topic. However, it became evident by the fifth interview that this approach was not needed; participants were all comfortable

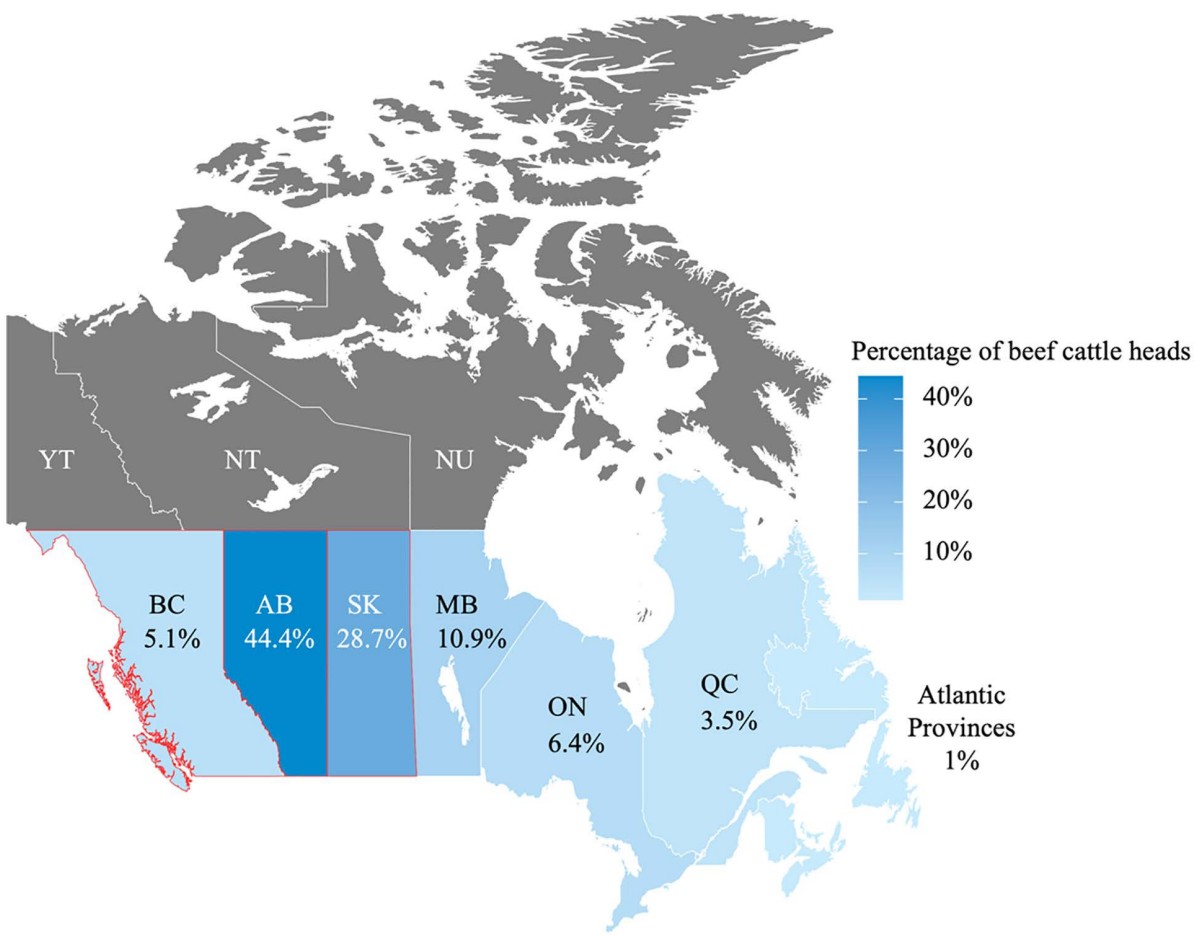

**Fig 1. The distribution (%) of Canada's 3.4 million beef cattle across Canada (as of January 2025).** Provinces where study participants were located are outlined in red. The color gradient indicates the proportion of the national beef cattle herd, with darker shades representing higher concentrations. Data source: [26]. Analyses and maps were generated using R (https://www.r-project.org/).

directly discussing the role of beef-on-dairy breeding strategies in dairy-beef production systems regardless of their specific knowledge about it. The questions about cull cows and veal calves did not contribute to addressing the research question of this study and were removed from the interview guide for the remaining participants.

The order of questions in the interview guide was developed to facilitate a natural flow to the interview, beginning with the introduction of dairy-beef production broadly and then narrowing it to specific topics regarding beef-on-dairy animals. To begin, participants were asked to describe the relationship between the dairy and beef industries in Canada based on their own experiences. Participants were then provided the information that 20% of beef produced in Canada comes from dairy farms and asked about their awareness and thoughts about this fact. The interviewer then introduced the topic of dairy-beef production by briefly explaining why some dairy animals are supplied to beef and introduced the concept of beef-on-dairy animals. To account for the differences in knowledge about beef-on-dairy animals among participants, the interviewer asked participants how much they had heard about beef-on-dairy animals and whether they had any experience with rearing non-replacement dairy calves in their beef operation. In cases where participants indicated low awareness about this practice, the interview guide included information paragraphs explaining what non-replacement dairy calves are and the reasoning for dairy farmers to use beef genetics in their herds. Participants were then asked about their awareness and opinions about the use of beef genetics in dairy farms, the challenges of rearing beef-on-dairy calves, and their possible impact on the beef market. The interviewer prompted participants to discuss two specific topics related to dairy-beef production if these were not mentioned spontaneously: public/consumer views and cow-calf separation. These topics were selected because they represent key components of the ongoing discussions surrounding dairy-beef production [3].

Lastly, participants were asked to share their perspectives on the future of dairy-beef production. Specifically, they were encouraged to reflect on who should be involved in decision-making regarding beef-on-dairy breeding strategies and consider its potential long-term impacts on dairy-beef production and the overall beef market. Additionally, participants were invited to discuss any other topics or ideas related to dairy-beef production that had not been addressed during the interview. The complete interview guide is available in the S1 File (https://doi.org/10.5683/SP3/SD4SSJ). All interviews were conducted by the first author (BV), an experienced interviewer who employed efforts to build rapport with participants before interviews began.

## Data analysis

The interviews were audio recorded and transcribed verbatim using an online transcription service (Otter.ai). The transcriptions were reviewed for errors, anonymized and randomly assigned an identification number (e.g., P43). Participants were emailed a copy of their transcript and given the opportunity to edit the content prior to the analysis; only one participant asked for minor modifications.

To our knowledge, ours is the first study to use semi-structured interviews to explore beef farmers' perspectives on dairy-beef production in North America. Given the novel nature of the study, interview transcripts were analyzed using inductive thematic analysis [28,29]. This data-driven approach enables the analysis of interview data without preconceived hypotheses, ensuring that the emerging themes and codes are grounded in the data. The first and last authors carried out the initial development of the codebook. The first author, who conducted all interviews, familiarized herself with the data, while the last author provided an external perspective, becoming familiarized with the data through extensive reading of the transcripts. Together, these two authors collaboratively developed the first version of the codebook, sharing insights and resolving discrepancies through discussion. The codebook was then reviewed and finalized by all co-authors, who reached a consensus on the final codes and themes (S2 Table; https://doi.org/10.5683/SP3/SD4SSJ). Following codebook finalization, the first author and a research assistant collaboratively coded the transcripts to identify quotes to report the results. Interview quotes are included in the results section to illustrate themes, with edits made for clarity, and interview transcriptions are available in the supplementary materials following participants' consent to share anonymized data (S3 Files; https://doi.org/10.5683/SP3/SD4SSJ).

## Positionality statement

In qualitative research, the researchers act as data collection and analysis instruments, and their personal and professional experiences can influence the research process [30]. Therefore, the following positionality statements briefly describe the authors' experiences and backgrounds. BV is a female PhD student in the UBC Animal Welfare Program (AWP). She was born and raised in Brazil, where she earned her Bachelor of Science degree in veterinary medicine and her MSc in Animal Welfare. Although she did not grow up in a farming community, she gained practical experience at her partner's family beef farm in southern Brazil. Additionally, she acquired experience with dairy cattle by working at the UBC Dairy Education and Research Centre. DMW and MvK are professors at UBC and co-lead the UBC Animal Welfare Program, where they have conducted research on dairy cattle for more than 25 years, much of this funded by the cattle industry. MvK grew up on a beef cattle ranch in Canada and still has personal links with ranchers in western Canada.

## Results

A total of 20 beef farmers participated in the study. Two interviews included two participants from the same family (i.e., the total number of interviews was 18). Interview duration ranged from 21 to 94 minutes, with an average length of $44 \pm 15$ minutes (mean $\pm$ SD). Eleven and nine participants were male and female respectively, and 11 were from Alberta, eight from British Columbia, and one from Saskatchewan. Participant ages ranged from 20 to 71 years (averaging $53 \pm 16$ years); seven participants did not disclose their age. The sample included individuals engaged in various types of beef production, including cow-calf (breeding) operations, backgrounding (growing), and feedlots (finishing), with some participants involved in multiple types of production (e.g., both cow-calf and backgrounding; see S4 Table; https://doi.org/10.5683/SP3/SD4SSJ). Two participants worked in smaller businesses, selling primarily to close friends and family, but had experience with larger beef operations, and their comments were consistent with those of other participants. Some participants also had experience with dairy production, either through extended family or friends who operated dairy farms or by having previously worked on dairy farms.

Thematic analysis of the transcripts yielded three main themes: 1) the dairy and beef relationship, 2) attitudes to beef-on-dairy animals, and 3) a shared future (Fig 2). The first two themes were further divided into three subthemes, detailed in the following sections.

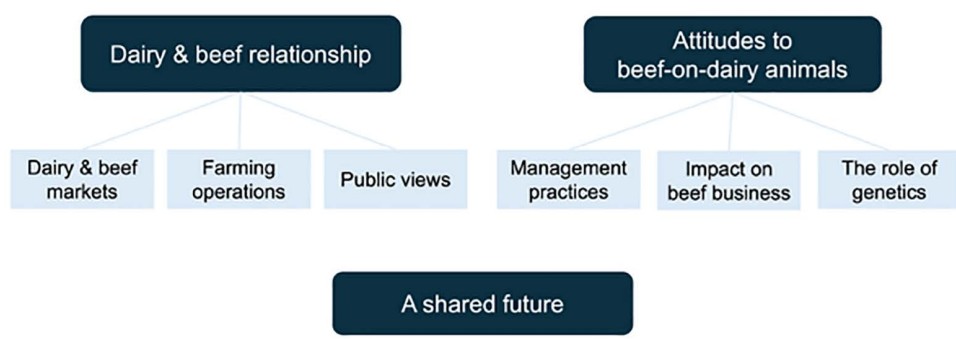

**Fig 2. Thematic map depicting the themes and subthemes from semi-structured interviews with 20 Western Canadian beef farmers on their views about dairy-beef production.** Black boxes represent themes, and white boxes represent sub-themes, which are connected with lines to the main themes.

**Dairy & beef relationship**

In this theme, participants discussed the relationship between the dairy and beef industries, emphasizing how market structures influence this relationship, how their farming systems differ, and how these differences shape public perception of the two industries.

**Dairy & beef markets.** The relationship between the Canadian beef and dairy industries was considered to be influenced by their commercial ties. Participants described this relationship as symbiotic because "*dairy cows are […] part of our whole beef industry*" (P56). However, when the interviewer shared that 20% of the beef produced in Canada comes from dairy operations [16], participants showed mixed responses. Some shared that they were surprised as they either did not "*think […] it was that high*" (P59) or thought "*it's higher than that*" (P89), and some envisioned that this percentage would increase because "*a lot more dairies are obviously using beef [genetics] to breed their cows*" (P94).

A common belief among participants was that there were differences in income streams between the beef and dairy industries, largely because Canada has a supply management system for dairy that allocates quotas to match production with domestic demand, helping to ensure stable prices and farmer income. The lack of financial certainty in the open beef market was considered a disadvantage in comparison to the dairy supply management system, as it meant that the beef business was "*subject to market forces*" (P33) while dairy farmers have "*a steady income*" (P29). Dairy supply management was perceived to influence Canada's international trade negatively, which in turn could impact the beef market and was viewed to cause "*some animosity between the two [industries]*" (P94). One participant perceived this to be counter-productive because "*they [dairy and beef industries] tend to lobby against each other's products when ultimately, they are the same product [as] every dairy animal becomes a beef product*" (P40).

**Farming operations.** Participants discussed the similarities and differences between dairy and beef farming operations. While some participants recognized that the two industries share certain similarities because "*at the end of the day, they're all cows, right?*" (P4), others perceived them to be "*very different [because] they operate in very different manners*" (P77). One key difference noted by participants was that "*compared to a dairy farmer, […] the beef industry is more traditional [and] not as motivated to change; whereas, the dairy industry [is] on top of technology* (P29).

Participants described the methods of raising animals as very different between dairy and beef operations. For example, "*a dairy farmer does everything they can to get that calf away as soon as they can. A beef farmer will do everything they can to get the cow-calf pair to work together*" (P13). Participants frequently characterized dairy farm operations as intensive because animals are confined in barns; whereas, their own beef operations were described as extensive and more aligned with the animals' natural environment. These differences in rearing systems also led some to state that they felt that this impacted the animal's quality of life. For example, "*our animals aren't raised in a confined situation, they're not raised in barns, we don't have the disease and the animal welfare issues that come with confinement. Our animals are largely raised outdoors on pasture in kind of a natural environment almost more like wildlife*" (P33).

However, some participants commented on the "*similarit[ies] [between dairy and beef operations] in what they call a confined feeding operation. In the dairy industry, […] all the cows are contained within that barn, similar to the […] majority of the [beef] animals [that] are raised in a feedlot as opposed to being out on a pasture or anything like that*" (P77). However, another participant argued that the feedlot phase represented only a short period of time in the beef animal's life as it was "*maybe 20% of their life [that] they're going to spend in a feedlot, the rest of the time they're going to be out in a natural environment romping around and having it pretty good so I think that's a good thing. [In contrast, the dairy animals] will never leave that sort of intensive type setting, for their whole life*" (P33) and instead are "*on basically concrete*" (P77).

**Public views.** Participants recognized the challenges related to the public views of farming, rooted perhaps in the perception that "*there are people who are generations removed from agriculture*" (P29) and "*people from the city do not understand what happens on the farm*" (P79). Both dairy and beef industries were perceived by some of our participants to be subject to negative public views, albeit sometimes for different reasons, as explained by one participant:

*"I think they both get a bad rep[utation] for sure. But in a different way. Like I see a lot of criticism in the dairy industry about taking the calves away from the parents […], versus the beef industry […], if you're looking at more feedlots, people are saying, 'Oh, they're crammed in, oh, they're eating this horrible food'. There's just a lot of misinformation. […] So, I think there's kind of different criticisms based on [each industry's practices]."* (P4)

Other participants believed that *"the dairy industry is probably under more scrutiny than the beef industry [because] you don't see dairy cattle out on big open pastures. If you're coming from a city and you don't know how the beef industry or dairy industry works. I can see how you would view the dairy cows' lives as being very poor"* (P29). In comparison, beef cows are *"basically raised for food production out in a natural environment. They can't really be criticized [as being] factory farm animals"* (P33). Some participants clarified that they did not believe that dairy cows have a poor life, but others said that they did not agree with some of the common dairy practices. This split sentiment was conveyed by two individuals, the first stating that *"those cows have very strict health protocols and programs; they have everything that they would ever need, right?"* (P29), while the other stated:

*"[people] don't see [any cows] outside. And I don't particularly like that. […] I'm not saying that they're not very well cared for or very healthy, but I would rather see them get a chance to go outside a little bit more. […] Whether it's right or wrong, I am not arguing that. It's not how I would do it"* (P31).

Participants mentioned the importance of the dairy industry communicating and being transparent with the public, because *"the public [doesn't] like to hear that they're euthanizing calves that they don't need, or that they aren't getting the very best of care […]. as the consumers [are] the ultimate purchasers of our product"* (P56). Participants also identified challenges regarding transparency with the public in indoor 'confined' dairy systems. As explained by one participant:

*"People drive by our farm all the time in the spring when we're calving, and they just love to pull over and watch the little calves out there with their moms, right? [But when] you drive by a dairy [farm], you don't even know if it's a dairy farm, a chicken farm, or a pig farm because you don't see anything. […] I think it almost makes it look like you're trying to hide something, and I know they're not. But I think people from the city like to drive around and see, they like to know where their food is coming from. So, like we say, […] we are not hiding anything.* (P31)

One participant also discussed the role of the Internet and the media in shaping the public views about cattle farms:

*"If you look at the cases where there's any kind of media coverage on cruelty to animals or whatever, almost always, it's a black and white dairy animal that's involved. Right? And so, I think the point is, if you're a consumer, and you're not at all familiar with ranching or the beef production system, [what] you would take away from that [is that] all cattle are treated the same. And that is not the case; our cattle graze out, and they have a much better life than other cows"* (P43).

### Attitudes to beef-on-dairy animals

In this theme, participants shared their perspectives on dairy-beef production and associated management practices, considering the potential impact on the beef business and the genetics of beef-on-dairy calves compared to pure bred dairy and beef calves.

**Impacts on beef businesses.** Participants talked about the production of dairy-beef and how it could affect the beef market. Some participants were aware of the challenge facing the dairy industry regarding non-replacement dairy calves, particularly in terms of the economic impact, stating for example: *"Twenty years ago, if you had a bull calf in a dairy, you*

had problems getting rid of it" (P13) because "there was no market for them" (P79). As a result, "they [dairy farmers] used to just kill the bull [calves]" (P43), and that "is actually pretty sad" (P13). This situation was regarded as "a real problem for the dairy industry" (P56) and thus used as a rationale to explain, and arguably support, why dairy farmers were incorporating beef genetics into their herds, such that participants would not "resent the dairy farmers for trying to get more value-added off their surplus calves" (P56).

Participants shared positive views about beef-on-dairy animals, describing them as "an opportunity [because dairy calves] are going to be coming into the beef industry anyways, which if there's a gap, it's going to get filled" (P40)". Some participants conveyed that they did not believe that beef-on-dairy animals would "affect the price of beef" (P13), but others were more skeptical and considered it to be beneficial only "as long as it doesn't oversaturate the market, and then be a detriment to just primary beef producers" (P4) as "it could drop [the beef prices] a little bit, right?" (P29). One participant noted that the instability of the beef market would be a key influencer:

"Well, at a time like this, where the [national beef] herd is shrinking, and [the beef industry is] short of animals, I think that everybody's going to perceive it as a good thing. If our [national beef] herd increases, again, [which will happen as we are a] very cyclic industry […] then [beef farmers] can start looking at it and say [that] the dairies are over-supplying […] the market." (P89)

Participants believed that dairy-beef production may impact different segments of the beef supply chain in different ways, potentially benefiting feedlots but compromising cow-calf operators. They noted, "By using dairy cattle or dairy crosses, [feedlots] can fill those pens and keep their operations working more efficiently" (P89). However, this may pose a threat to cow-calf operators, who worry that "their calves are going to be worth less if they're competing against a […] cull product like beef on dairy" (P40). Participants also shared their perspectives and predictions about the potential impact of dairy-beef production on beef farming in the context of the traditional methods of beef production and the overall trajectory of the beef industry. Participants were concerned that the increase of dairy supply to beef would mean that "the dairy-beef is just going to wipe us out, it's going to happen slowly, but [its] going to happen" (P33), "[…] so you might see less and less actual beef production as a result of the beef on dairy" (P4).

Participants also discussed the possible impacts that dairy-beef production might have on intensifying beef operations. Some participants were concerned that the beef farms and ranches would need to modify their practices to remain competitive with dairy-beef production because "the reality is, [dairy farmers] can probably put out a pound of beef cheaper than I can. […] I could become more intensive, but then my operation would start to look more and more like a dairy farm. […] But all of those things, kind of push you from the traditional beef industry" (P33). Participants anticipated that the beef industry would experience further consolidation, as "you're going to see fewer [bigger] farms. I hope that we still get to see the family farms, [but] I think a lot of them are going to be corporate" (P77), which was perceived to be a threat to "the art part of raising livestock" (P33) because "You [won't have] that personal touch like you used to have" (P79).

One participant also expressed concerns about the potential impact of dairy-beef production on public perception of beef farming and its products, because "If people can't distinguish [dairy-beef and pure beef products], then that may be something that would deter them from consuming the beef, because if they hear that the dairy ones aren't raised in a humane way […], then that may inhibit their consumption of our product, which is beef" (P56). This sentiment was used by some to highlight that beef producers "[should] hang on to the fact that their animals are out on the range and harvesting sunshine to make good beef. […] Our animals have a good path as part of their life, have access to free range and that they are in what people like to think the traditional settings are" (P56).

When reflecting on the overall views of beef producers regarding dairy-beef production, one participant mentioned having heard their peers express a range of opinions – from positive sentiments like "[dairy beef] is a great opportunity" to negative perspectives such as "[dairy beef] is a threat to our industry, and those calves will compete with ours" (P51).

Another participant had never thought about dairy-beef production before the interview and believed that others might also lack awareness about this practice, but once aware, expressed that they "*have worries […].[And] I hope that [it] is not a problem for you when you present your data that ranchers that you spoke to weren't even aware of what the dairy people were up to. And now that they are, they're worried and annoyed*" (P70).

**The role of genetics.**  Participants discussed the role of beef and dairy genetics in the suitability of calves for meat production. Some participants were optimistic about increased adoption of beef-on-dairy breeding strategies because "*if [non-replacement dairy calves] weren't beef cross, they would be straight Holstein or straight Jersey, which are less economical for guys like us to feed and convert into beef*" (P31). Holstein cattle were perceived to have "*more bone in the carcass*" (P13) but were also known to produce "*very good beef once it's grain-fed and fattened up*" (P59) and that "*the meat could taste just as good as any beef. But it's not cost-effective*" (P79). Participants highlighted the potential benefits of crossbreeding dairy and beef to enhance Holsteins' capacity for beef production, stating that "*if they breed in some really good beef genetics, those cattle would be more efficient, which would make it more attractive for us to bid on them*" (P95). However, dairy-beef calves were still seen by some of our participants as having lower meat quality than purebred beef calves: "*if you're looking for steak and whatnot, and someone wants a marbled steak, they're not going to buy anything that's got Holstein in it*" (P13) and thus some participants felt that this product was not as "*marketable as your prime beef breeds like Angus*" (P77).

Some participants believed that processors and consumers might not notice or care about the difference between crossbred and purebred beef products. One participant stated, "*I don't think you can tell if you go to the store and buy beef, whether that was dairy or ranch beef. You may [see that] they aren't getting graded AAA ribeye steaks and things; those are probably beef, not dairy. But no, I don't think the public can tell*" (P51). Another participant added, "*The processors really don't care [if those animals] are dairy or beef. It's what standard they get and what they're going to do with it. So those canner cows and dairy products? That's our McDonald's beef, and there's nothing wrong with it*" (P93).

Although some participants believed that "*genetics is what it comes down to [when] getting a good dairy beef cross*" (P95), others mentioned the importance of the "*circumstances [in which] that [dairy] beef calf is raised. Because they will be raised in a similar situation as the other dairy calves and will also be subjected to the same stresses*" (P56). This latter point was perceived to influence the quality of the final product, as one participant argued that "*Selfishly, I would be able to distinguish that the meat from my animals was not raised that way*" (P51). Another participant added, "*I don't think the quality of the beef would be quite the same. I mean, I'm saying that from a prejudiced point of view, but one of the ways that you get your top grades is from muscle. [If] the calves are raised from birth in fairly confined quarters, they won't have the same muscle that the animals that have been out on range*" (P56).

Participants also shared that "*one of the challenges with [Holstein calves] is [that] they often get sick and die*" (P13), but the genetics of beef-on-dairy calves were perceived to enhance the resilience of the animals. As one participant noted: "*I do find there is a higher death loss with the Holstein, […] and then the crosses are a little bit lower than the straight Holsteins but still higher than all beef cattle*" (P95). The resilience of animals to environmental conditions was believed to play a key role in their survivability because "*The beef-dairy cross cattle have a bit of beef in them, [which] makes them a bit hardier, and they can [better] withstand the elements*" (P31).

**Management practices.**  Participants shared their perspectives on the raising of non-replacement dairy calves, comparing it to the practices used for beef calves, discussing how these differences affect the performance and resilience of calves, and providing recommendations based on their experiences. Participants expressed that the rearing environment was as important as genetics because "*it's 50/50 on both [environment and genetics]*" (P31), so "*[dairy farmers] may be crossing [dairy cows] with a beef breed, but they're still raising the animals the way they would dairy breeds*" (P77). Participants discussed how the differences in management practices between beef and dairy operations would affect dairy-beef production, as one participant shared: "*My perception and I think most people in the beef industry's perception, would be that those surplus calves are still being raised pretty much in confined quarters and [under] more*

*intense, more concentrated circumstances than most beef cattle are*" (P56). Another participant also criticized the rearing of calves destined to beef production using dairy farming practices because "*those calves are an entirely different production system in themselves, and they need to be managed that way […]. I think the mistake is made by trying to slot them into an existing system, as opposed to just defining them as what they are and writing the optimal approach*" (P51).

Participants' concerns regarding calf-rearing practices were linked to specific dairy practices, including indoor confinement and the early separation of calves from their mothers. Participants perceived indoor confinement as negative to animal welfare because "*that's [not] their natural way to live*" (P70) and compared it with the beef calves, saying that they "*see [beef] calves […] running in groups of 5 and 10, and their tails are up in the air. They have fun. They're a bunch of babies having fun. […] So why should those beef dairy calves not be able to have the same thing?*" (P81). Some participants also believed that confinement compromised the immune system of animals; with one participant stating that "*we have a stronger immune system [in the beef industry] because they are out on pastures […], whereas a dairy calf is very confined*" (P29). Health and nutrition were considered most important to guarantee that "*dairy-beef calves are a valuable by-product coming off the [dairy] farm*" (P51). For animals to be healthy, participants believed that they should "*stay on the farm till two weeks old […] to give it a good start*" (P89). Participants believed that calves supplied from dairy to beef could face "*health issues because [when] adjusting an animal to a completely different environment you're always going to run into challenges*" (P4). Exposing animals to the outdoors was also considered to be important to "*[Expose them to] some different environmental factors […] instead of them being undercover the whole time*" (P29), and the use of social housing was seen as valuable for "*making sure that they can be in a group*" (P29) later in life.

Participants also believed that the common practice of early cow-calf separation in the dairy industry compromised calves' resilience because "*They separate them way too young and […] they're transporting them [when] they're just way too fragile*" (P13), and "*coming off their mum like that [is] a lot of stress*" (P95). Participants shared their experiences with keeping cows and calves together: "*Based on what I see in the cow-calf side of things, I think that calves do better and their health is better when they stay with their moms for longer. I see the need for [this] in the dairy industry*" (P4). Some participants did, however, acknowledge that dairy farmers need to separate the calf from the cow soon after birth because "*they have to do that in order to get the milk production, right? So, if you left the calf on too long, you're missing out on a lot of milk*" (P95). Some participants also struggled to propose solutions for dairy farmers because "*I don't know what the answer is. If there was enough incentive to say that [is] unacceptable, then maybe they do like we do here with our nurse dairy cows*" (P70).

Not surprisingly, some participants believed that public perception about some of the dairy farming practices could jeopardize their image with the public. When discussing rearing practices, participants believed public values should be taken into account because "*The public has to be able to see and accept it. [And the public] would not like the idea that a calf is taken away from [their] mother on the first day and then put in an individual hutch*" (P70). One participant voiced that dairy-beef calves "*spend their entire lives from the time they're born, until the time they are slaughtered in a confined factory operation […]. [People] view factory farms as something that's wrong with agriculture, not something that's right with agriculture. […] [So] do we want to be associated with factory farms?*" (P40). Participants pondered the risk of dairy-beef production to the public image of beef farming, as explained by another participant: "*Implementing dairy into a beef scenario? Personally, I wouldn't really want the public to know that, to be honest with you. [Because] I think if the public found out that dairy cattle are then being used for beef production, I just think it would [be] opening up a whole other can of worms*" (P29).

### A shared future

In this theme, participants reflected on the future of dairy-beef production and discussed who ought to be involved in shaping the production system. Participants felt that the dairy industry ought to take the lead in discussions regarding the future of dairy-beef production, but the beef industry should also be involved because "*it shouldn't be a one-sided thing. I*

would say the beef and dairy [industries] need to coordinate together and talk about it and collaborate [to] decide what's going to work best for both industries" (P33). Participants also noted that they "don't think there needs to be an agreement but, as things move forward, there needs to be [...] continued dialogue [between industries because] we're not competitors [and] we should be working in synergy" (P93). Including different actors along the beef supply chain in these conversations was considered important for achieving a sustainable future in dairy-beef production. One participant articulated their views on this matter as follows:

> "You would have to have all aspects of beef production [involved] too. You wouldn't just have a feedlot. You need cow-calf producers, feedlot producers, packing plants, heck, even grocery. The whole food chain of beef production, I think should be there [because] every group [within the] beef production [value chain] has a different viewpoint on it." (P29)

Participants also predicted some challenges in the communication between the two industries "because [...] I don't know [if] there's a lot of willingness or interest in cooperating" (P33). One participant proposed offering evidence to support suggested changes as a way to engage with people who may not be willing to collaborate: "Being honest, I think that if either industry was going to say, this is a good thing, then they need to come out with some numbers to support that, whether it's the feeding industry or the beef industry in general. [...] That's the way I would approach that conversation [because] people don't like being told things [...], so give them the information to believe it" (P40).

Participants also viewed beef-on-dairy animals as an opportunity for both industries to collaborate to "turn [a surplus animal] into a viable food product with [...] a minimal footprint, [...] best not only for animal welfare, but also for animal health, food safety, lots of different aspects" (P51). These benefits were seen to strengthen the relationship between beef and dairy because "These are ways that we can integrate and benefit from each other" (P93). However, some viewed this only as possible if "the two industries can manage to get over or agree to disagree on differences in supply and non-supply managed industries; that it could be a very copacetic relationship that benefits all of us, or if they can't, it might not." (P40). Communication and collaboration between the two industries were perceived as helping both industries benefit from dairy-beef production because "It's coming, whether we like it or not; it's how do we work with it?" (P81)

Participants also noted the importance of collaboration in managing calves across the two industries, as there needs to be "a very close communication between the dairy operation and the operation that will be receiving those calves [...] so that none of those pieces [such as nutrition, castration, vaccination] fall between the cracks, more than about saying this is the right way and this is the wrong way" (P51).

## Discussion

### Beef-on-dairy: Opportunity or threat?

Participants expressed mixed feelings about beef-on-dairy breeding, with some viewing it as an opportunity to mitigate the declining beef cattle numbers and others viewing it as a potential threat to cow-calf operations if market saturation occurs. The total number of cattle in Canada has declined; the national herd size in January 2025 was at its lowest since 1988 [26]. Beef cattle numbers fell from nearly 13 million in 2005 to about 9 million in 2025, while dairy cattle have remained stable at approximately 2 million [26]. Unlike beef calves from cow-calf operations that are typically born in the spring and enter the market primarily in the fall, dairy-beef calves are produced year-round, offering a consistent supply of animals into the feedlot, which could help stabilize the beef inventory. Projections for 2026 suggest that up to 6 million dairy-beef calves will be born in Canada and the U.S. [31]. However, the extent to which these concerns are valid is uncertain, as it remains unclear how many of these animals will ultimately enter the beef supply or how they will compete in the market alongside conventionally raised beef cattle. For example, while some participants expressed concern about an increase in the number of dairy-origin calves entering the beef sector, others believed the overall number of calves would remain stable, with the primary change being the use of crossbred dairy-beef genetics. Nevertheless, our study indicates that

Canadian beef farmers share concerns that dairy-beef calves could create competition, potentially lowering calf prices for cow–calf producers.

Most participants viewed beef-on-dairy calves as less desirable than purebred beef calves, with the latter believed to command higher prices at auction [32], and considered that the benefits of beef-on-dairy animals primarily stemmed from superior meat compared to that from purebred dairy calves. There is some evidence that beef-on-dairy calves have higher carcass weights and better conformation than purebred dairy calves but remain inferior to purebred beef calves [33,34], aligning with participants' perspectives. However, there is only a dearth of information on performance comparisons between dairy and beef-on-dairy calves entering the beef market [35], and findings vary depending on the type of Holstein genetics (e.g., New Zealand vs. North American) and the genetic merit of beef sires [36].

Despite differences in meat production traits between dairy and beef breeds, participants believed that consumers might not distinguish between beef-on-dairy calves and conventional beef products. Research supports this, showing few differences in tasting panel acceptance of steaks from conventional beef cattle, beef-on-dairy cattle, and pure dairy cattle [37]. Interestingly, the same study found that a trained panel rated strip loin steaks from dairy and beef-on-dairy cattle as having a more tender and intense, buttery fat flavour than conventional beef steaks. Beyond taste, appearance is important to retailers as color stability and steak shape can affect marketability [38]. Dairy breed steaks discolor more quickly due to higher oxidative muscle fibres and tend to be smaller and more angular than beef breed steaks [37]. Yet, other studies show no differences in these traits between beef-on-dairy and pure beef loin or ribeye steaks [37,39]. Advances in knowledge and perimortem protocols may further minimize breed-related differences in meat products, particularly in processed meats [35], supporting the view that consumers and retailers may not distinguish dairy-beef from conventional beef.

Beyond genetics, early-life experiences have lasting effects on carcass weight and conformation, with beef-farm-raised calves usually outperforming calves from dairy farms, regardless of genetics [40]. While introducing beef genetics can improve beef-related traits in calves destined for meat production, further research is needed to understand how beef-on-dairy calves compare to conventional beef cattle, particularly regarding genetic and environmental interactions.

## Raising beef-on-dairy calves

Participants believed that beef cows experience better welfare than dairy cows; this perspective was shared by a panel of 70 cattle experts who assessed the likelihood of cattle experiencing negative welfare states as being higher in dairy than beef systems [41]. Given this perception of higher welfare for beef cattle, participants worried that the blending of meat from beef and dairy production would undermine consumer trust in the product, potentially impacting the beef industry. Previous work suggests that animal welfare concerns influence meat consumption more than dairy, as most vegetarians still consume dairy products [42], and animal welfare concerns are a primary reason for reducing meat intake [43]. However, participants' worries that public awareness of dairy farming practices could negatively impact the beef marketing are warranted. Several studies indicate that the public values practices that are not common in North American dairy operations, such as pasture access and cow-calf contact [14,44,45]. The extent to which the public becomes aware of the dairy origin of meat remains unknown, as does the potential impact of such awareness on the public image of beef farmers. Nevertheless, beef farmers expressed concerns about the risks this practice may pose to their industry, given the substantial differences between beef and dairy production systems, farmers' lifestyles, and the role of tradition as a core value among beef producers [46].

As noted by participants, the public often learns about contentious farming practices through undercover videos and media exposés [47]. Such cases can influence the public image of animal production, reducing trust in farming [48]. Aligning farming practices with public values to support the social sustainability of the cattle industries was deemed important by participants, who saw the public being able to see beef cows and calves on pasture [46] as beneficial in building trust [49]. Participants also believed that beef-on-dairy calves reared using typical dairy calf care conditions (e.g., indoors,

individually housed, without dam contact) would compromise calf health, welfare, and resilience. In contrast, dairy farmers often justify these practices for their perceived benefits to animal health and welfare [50–52]. For example, dairy farmers express concerns about cow-calf contact systems, citing potential issues such as inadequate colostrum intake, increased stress from delayed separation, and lack of shelter for calves outdoors [53]. In contrast, our participants viewed cow-calf contact as beneficial for animal welfare and calf resilience. Multiple studies have examined the impact of cow-calf contact on the health and welfare of dairy cows and calves (see reviews [54,55]), the results of which mostly align with the perspectives of the participants in this study that the practice of separation may compromise calf health and welfare. Despite this, our participants also acknowledged that integrating prolonged cow-calf contact into existing dairy milking systems presents logistical challenges, aligning with dairy farmers' views [53], and consistent with a type of technological 'lock-in' that makes it difficult for systems to escape specific historical paths [56].

Participants provided recommendations for dairy farmers based on their cattle-raising experience, suggesting that practices such as outdoor access and social housing would improve calf adaptability, enhancing welfare and resilience. Consistent with participant views, multiple studies have identified the benefits of outdoor access and social housing for dairy cattle [57–59]. Participants also recommended extending the period that dairy-beef calves remain on dairy farms to foster resilience before subjecting calves to transport. Historically, non-replacement dairy calves typically leave the home dairy farm between 3 and 7 days of age, and sometimes even within 1 day of birth [60]. Aligning with participant concerns, early-life transport negatively impacts calf performance [61] and compromises calf health [5]. The Canadian Cattle Transport Regulations [62] now require that calves must be at least 8 days old before transport. However, as participants suggested, prolonging the time calves stay on the home dairy farm may not be sufficient without other changes in practice that improve their welfare, health, and resilience.

### Towards a collaborative and sustainable future

Participants emphasized that the overall impact of dairy-beef production on the beef industry would depend on the quality of collaboration between the dairy and beef sectors toward shared goals. Dairy-beef production was viewed as potentially benefiting both industries, reducing the environmental footprint of calf production, improving the welfare of non-replacement dairy calves, and enhancing the genetic traits of calves entering the beef supply from dairy herds [11,35,63]. Input from the beef sector was considered crucial for achieving these outcomes while mitigating potential negative effects on beef operations.

Participants believed that discussions about the future of dairy-beef production must involve various groups within the supply chain, including cow-calf operations, feedlots, processors, and retailers, as each group was thought to bring valued perspectives. Representatives from post-farm gate organizations in Australia also underscored the necessity for collaboration, leadership, and commitment across sectors in addressing the management of non-replacement dairy calves [3]. Our participants displayed varying knowledge of dairy-beef production, with some well-versed in the practice and others learning about this for the first time during our interviews. This variation in knowledge suggests that not all sectors are equally engaged, which may pose a challenge to promoting a sustainable future for dairy-beef production. These findings underscore the importance of enhanced communication and knowledge exchange between the dairy and beef sectors to ensure that producers are equipped with accurate information about implications of beef-on-dairy production.

Participants pointed out challenges that could hinder the collaboration between the dairy and beef industries, including Canada's dairy supply management system [64]. While this system helps the dairy sector, it was seen as a source of conflict between the industries [65], complicating the market dynamics and potentially making collaboration more difficult. Despite differences between the industries, Canadian beef and dairy farmers hold shared values regarding animal welfare and their duty of care toward animals [46,51], and this could provide common ground for discussions about the future of dairy-beef production.

Our results emphasize the potential for dairy farmers to raise non-replacement calves as contributors to the beef supply chain. Reframing the dairy industry's narrative to view beef-on-dairy calves as an opportunity rather than a byproduct could be key to promoting better calf care management practices. The results also suggest a need for enhanced communication between the dairy and beef industries to support the development of best practices for calf care. Progress in this area will likely depend on the emergence of leadership to establish and promote standards of care for beef-on-dairy caves. While our participants believed that the dairy industry should be responsible for this leadership, they emphasized the importance of involvement from multiple sectors within the beef industry. Further research is needed to explore how such communication and coordination might develop, including what strategies could engage the various sectors involved.

## Limitations and future research

It is important to acknowledge the limitations of this study. First, the use of convenience and snowball sampling likely introduced selection bias. Although we attempted to engage national and provincial beef producer organizations to diversify recruitment, these efforts were unsuccessful. Future studies may benefit from establishing stronger collaborations with these organizations, which could facilitate broader participation among farmers. Participatory research methods, where knowledge users are actively involved in multiple stages of the research process [2], may also increase engagement and uptake by farmers.

Second, interviews were conducted by phone. This method allowed participation from farmers across Canada, including those in remote areas, but limited the ability to observe non-verbal cues and may have reduced interpersonal connection. Future studies could consider using in-person interviews to foster more engaged dialogue. Conducting interviews at industry events may be a practical way to increase access to in-person interviews while also reaching geographically dispersed participants, a strategy that has proven effective in other studies [e.g., 66].

Third, while one-on-one interviews allowed participants to express their views freely, they did not offer the opportunity to observe how opinions are shaped through dialogue, as would be possible in focus groups. Our findings illustrate variation in perspectives between beef and dairy producers and among beef farmers. Focus groups involving either multiple beef farmers or both beef and dairy farmers could offer insights into barriers to communication and collaboration that participants identified as challenges for the future of beef-on-dairy production.

Furthermore, interview studies are inherently limited in their ability to collect data from a broader sample of the population. Future research could adopt a mixed-methods approach, combining in-depth qualitative methods (e.g., interviews and focus groups) with large-scale surveys to capture a broader and more representative range of perspectives. This would also allow for the exploration of how attitudes may vary across regions.

Lastly, although our study focused on Canadian farmers and revealed context-specific considerations, such as the role of dairy supply management in the relationship between the dairy and beef industries, this topic is of growing global relevance. To our knowledge, the only other study addressing beef farmers' views on this topic was conducted in Ireland [17]. More research is needed in other countries to capture a diversity of perspectives and account for differences in cultural, economic, and policy contexts.

## Conclusion

This study employed semi-structured interviews to explore beef farmers' views on dairy-beef production, focusing on beef-on-dairy breeding to gain a deeper understanding of their perspectives on this practice. Participants' attitudes toward dairy-beef production were mixed, shaped by the connection between the dairy and beef industries and the distinct social, market, and cultural contexts of the two sectors. While participants recognized potential benefits of beef-on-dairy breeding, particularly in improving carcass characteristics of calves from dairy farms, they also raised concerns about market competition and threats to traditional beef production practices. Including beef farmers' perspectives in discussions about the future of dairy-beef production was seen as essential to ensuring that this development benefits both industries.

## Supporting information

**S1 File. Semi-structured Interview Guide.** Interview guide used to explore beef farmers' views on dairy-beef production, including open-ended questions and follow-up prompts.
(PDF)

**S2 Table. Codebook.** List and definitions of themes and sub-themes developed through thematic analysis of the interview transcripts.
(PDF)

**S3 File. Interview Transcripts.** Anonymized transcripts of the interviews conducted with 20 beef farmers, used for thematic analysis.
(PDF)

**S4 Table. Number of Participants by Beef Operation Type.** Summary table showing the distribution of study participants across different types of beef production operations (e.g., cow-calf, backgrounding, feedlot).
(PDF)

## Acknowledgments

We thank the farmers who participated in our study. We also thank our research assistants, Mahshid (Michelle) Heydarirad and Shirley Yang, for their help with audio transcription and data coding, and Dr Sarah Bolton (Animal Welfare Program) for helpful comments on a previous draft of this manuscript. Please note that the funders had no role in study design, data collection and analysis, decision to publish, or preparation of the manuscript.

## Author contributions

**Conceptualization:** Bianca Vandresen, Marina A.G. von Keyserlingk.

**Data curation:** Bianca Vandresen, Marina A.G. von Keyserlingk.

**Formal analysis:** Bianca Vandresen, Marina A.G. von Keyserlingk.

**Funding acquisition:** Daniel M. Weary, Marina A.G. von Keyserlingk.

**Investigation:** Bianca Vandresen, Daniel M. Weary, Marina A.G. von Keyserlingk.

**Methodology:** Bianca Vandresen, Daniel M. Weary, Marina A.G. von Keyserlingk.

**Project administration:** Daniel M. Weary, Marina A.G. von Keyserlingk.

**Resources:** Daniel M. Weary, Marina A.G. von Keyserlingk.

**Supervision:** Marina A.G. von Keyserlingk.

**Visualization:** Bianca Vandresen.

**Writing – original draft:** Bianca Vandresen.

**Writing – review & editing:** Daniel M. Weary, Marina A.G. von Keyserlingk.

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
