## [Decision Letter · Decision Letter 0]

16 Jul 2025

PONE-D-25-22276It’s not black and white: Perspectives of Western Canadian beef farmers on dairy-beef production.PLOS ONE

Dear Dr. von Keyserlingk,

Thank you for submitting your manuscript to PLOS ONE. After careful consideration, we feel that it has merit but does not fully meet PLOS ONE’s publication criteria as it currently stands. Therefore, we invite you to submit a revised version of the manuscript that addresses the points raised during the review process.

We look forward to receiving your revised manuscript.

Kind regards,

Juan J Loor

Academic Editor

PLOS ONE

Journal Requirements:

Social Sciences and Humanities Research Council of Canada

Reviewers' comments:

Reviewer's Responses to Questions

**Comments to the Author**

1. Is the manuscript technically sound, and do the data support the conclusions?

Reviewer #1: Partly

2. Has the statistical analysis been performed appropriately and rigorously? 

Reviewer #1: N/A

3. Have the authors made all data underlying the findings in their manuscript fully available?

Reviewer #1: Yes

4. Is the manuscript presented in an intelligible fashion and written in standard English?

Reviewer #1: Yes

5. Review Comments to the Author

Reviewer #1: 1 – Please consider adding a map of Canada showing the regions where the research took place. This would help readers better understand the study's location and context.

2 – The practical reason for choosing this target population needs to be explained more clearly. Why were these participants selected, and why are they important for this topic?

3 – What was the specific criterion for reaching data saturation? In snowball sampling, recruitment usually continues until no new information appears. Was the number of 20 participants truly enough, or could it reflect that participants were very similar in their views?

4 – The discussion could go deeper. Even though the study is based on opinions, some statements from participants seem incorrect or misleading. The discussion is a good place to address or correct this kind of information.

5 – Even if the study is mostly qualitative, some numbers, percentages, charts or tables would make the results easier to follow and more engaging (especially for a broad audience interested in the topic such as dairy, beef, environment, market)

6 – Include more details about the study’s limitations, such as selection bias or limitations of the qualitative method itself.

7 – How can we be sure the participants gave their honest opinions and were not influenced by the way questions were asked? Is the interviewer experienced in interviewing?

8 – Conclusion (lines 652–653) The first lines of the conclusion say the method was “effective,” but the study does not seem to evaluate the method itself. I suggest rephrasing this part.

6. PLOS authors have the option to publish the peer review history of their article (what does this mean? ). If published, this will include your full peer review and any attached files.

**Do you want your identity to be public for this peer review?** For information about this choice, including consent withdrawal, please see our Privacy Policy .

Reviewer #1: No

---

## [Author Response · Author response to Decision Letter 1]

21 Jul 2025

It’s not black and white: Perspectives of Western Canadian beef farmers on dairy-beef production.

Bianca Vandresen1, Daniel M. Weary1, Marina A. G. von Keyserlingk1*

1Animal Welfare Program, Faculty of Land and Food Systems, The University of British Columbia, Vancouver, BC, Canada

--

Journal Requirements:

The manuscript and additional files have been reviewed to follow the formatting style requested.

All data used in the analysis are available as supplementary material at:

Vandresen, Bianca; Weary, Daniel M.; von Keyserlingk, Marina A.G., 2025, "It’s not black and white: Perspectives of Western Canadian beef farmers on dairy-beef production.", https://doi.org/10.5683/SP3/SD4SSJ, Borealis, V1

Social Sciences and Humanities Research Council of Canada

Lines highlighted in yellow to double-check lines before submitting

Response to Reviewers

Reviewer #1: 1 – Please consider adding a map of Canada showing the regions where the research took place. This would help readers better understand the study's location and context.

AU: Thank you for this helpful suggestion. We agree that a visual representation would support readers’ understanding of the study’s geographic context. In response, we have included a map of Canada showing the distribution of beef cattle heads by province and have highlighted the provinces where participants were located. This is now presented as Figure 1 in the revised manuscript.

2 – The practical reason for choosing this target population needs to be explained more clearly. Why were these participants selected, and why are they important for this topic?

AU: We appreciate the reviewer’s comment and have clarified the rationale for participant selection in the revised manuscript. Specifically, we recruited individuals who earn an income from beef production and reside in Canada, referred to throughout the paper as “beef farmers.” This population was chosen for two key reasons. First, we sought to investigate the views of beef farmers on beef-on-dairy production, which necessitates engaging those directly involved in beef production, as explained in the introduction. Second, participants were required to reside in Canada because views on animal production systems are shaped by local cultural and market contexts. Given the unique dynamics of the Canadian beef and dairy sectors, a Canadian sample was essential to ensure relevance and contextual accuracy. We have clarified this in the revised manuscript (line 132), which now reads: “Since local context can influence peoples’ views toward animal production systems [25], we recruited only from within Canada to reflect the culture and market dynamics specific to this region.” The participation of beef farmers from the western provinces of Canada reflects the outcome of recruitment efforts, which were employed to recruit participants from across Canada, but resulted in participants being drawn primarily from these regions. This is not unexpected, as the majority of beef cattle production in Canada is concentrated in the western provinces. As clarified in line 135: “Despite the efforts to include farmers from across Canada, including offering interviews in French to accommodate participants from francophone regions, the final sample consisted of participants from three western provinces: Alberta, British Columbia, and Saskatchewan, which are the provinces with the highest concentration of beef production in Canada (Fig 1) [26].

3 – What was the specific criterion for reaching data saturation? In snowball sampling, recruitment usually continues until no new information appears. Was the number of 20 participants truly enough, or could it reflect that participants were very similar in their views?

AU: Thank you for raising this important point. We acknowledge the ongoing debate in qualitative research regarding the use of data saturation as a criterion for determining sample size. As discussed in the manuscript (lines 111–118), saturation is increasingly critiqued as a post hoc rationale that may not align with the interpretative and context-dependent nature of qualitative inquiry. Therefore, our sample size was not determined based on a formal assessment of data saturation.

Instead, as noted in the manuscript (lines 127–140), sample size was guided by pragmatic considerations, including the exhaustion of recruitment strategies. Recruitment began on September 1, 2023, and continued until May 10, 2024, when no further prospective participants could be identified through snowball sampling.

We have added further clarification in the revised manuscript at line 127: “Given these limitations, our study used a snowball method to recruit a convenience sample of participants for phone interviews; recruitment ceased once all known referral paths had been exhausted.”

And at line 134: “Recruitment began on September 1, 2023, and ended on May 10, 2024. Despite the efforts to include farmers from across Canada, including offering interviews in French to accommodate participants from francophone regions, the final sample consisted of participants from three western provinces: Alberta, British Columbia, and Saskatchewan, which are the provinces with the highest concentration of beef production in Canada (Fig 1) [26].”

Additionally, the final sample of 20 participants aligns with sample sizes reported in other studies undertaken by other research groups employing similar qualitative methods (e.g., Shortall, 2022; Wemette et al., 2020), and while some shared views emerged, we also observed meaningful diversity in participants’ perspectives, as described in the results section.

Shortall, O.K., 2022. A Qualitative Study of Irish Dairy Farmer Values Relating to Sustainable Grass-Based Production Practices Using the Concept of ‘Good Farming.’ Sustainability 14, 6604. https://doi.org/10.3390/su14116604

Wemette, M., Safi, A.G., Beauvais, W., Ceres, K., Shapiro, M., Moroni, P., Welcome, F.L., Ivanek, R., 2020. New York State dairy farmers’ perceptions of antibiotic use and resistance: A qualitative interview study. PLOS ONE 15, e0232937. https://doi.org/10.1371/journal.pone.0232937

4 – The discussion could go deeper. Even though the study is based on opinions, some statements from participants appear to be incorrect or misleading. The discussion is a good place to address or correct this kind of information.

AU: We agree that the discussion offers an opportunity to critically engage with participant statements that may appear inaccurate or inconsistent with current scientific or industry knowledge. As such, we have expanded the discussion in the following lines:

Line 561: “However, the extent to which these concerns are valid is uncertain, as it remains unclear how many of these animals will ultimately enter the beef supply or how they will compete in the market alongside conventionally raised beef cattle. For example, while some participants expressed concern about an increase in the number of dairy-origin calves entering the beef sector, others believed the overall number of calves would remain stable, with the primary change being the use of crossbred dairy-beef genetics. Nevertheless, our study indicates that Canadian beef farmers share concerns that dairy-beef calves could create competition, potentially lowering calf prices for cow–calf producers.”

Line 608: “The extent to which the public becomes aware of the dairy origin of meat remains unknown, as does the potential impact of such awareness on the public image of beef farmers. Nevertheless, beef farmers expressed concerns about the risks this practice may pose to their industry, given the substantial differences between beef and dairy production systems, farmers’ lifestyles, and the role of tradition as a core value among beef producers [46].”

However, we would like to emphasize that, in qualitative research, the accuracy of participants’ claims does not diminish their significance. These views, whether factually correct or not, are meaningful in shaping attitudes, decisions, and behaviours, and thus have real-world implications for the adoption and social dynamics of beef-on-dairy production. In fact, the presence of misunderstandings or knowledge gaps highlights the importance of enhancing communication and knowledge exchange among stakeholders. We have made this point more explicit in the revised manuscript (line 660): “These findings underscore the importance of enhanced communication and knowledge exchange between the dairy and beef sectors to ensure that producers are equipped with accurate information about implications of beef-on-dairy production.”

5 – Even if the study is mostly qualitative, some numbers, percentages, charts or tables would make the results easier to follow and more engaging (especially for a broad audience interested in the topic such as dairy, beef, environment, market)

AU: Thank you for this thoughtful suggestion. While we recognize the potential value of incorporating quantitative summaries to enhance accessibility, we respectfully disagree that this approach would benefit the current study. As this research was designed and conducted using only qualitative methodology, specifically reflexive thematic analysis, introducing numerical representations of codes or sub-themes would contradict the epistemological foundations of our analysis.

As discussed by proponents of thematic analysis (e.g., Braun and Clarke, 2006), quantifying qualitative data can inadvertently imply that frequency equates to importance, which risks oversimplifying the nuanced and context-dependent nature of participants’ values and perspectives. For example, a theme expressed less frequently may still hold significant conceptual weight in understanding participants’ positions on dairy beef production.

That said, we have carefully structured the results using clear subheadings, illustrative quotes, and thematic summaries to aid comprehension and engagement for a broad audience. We believe this approach maintains methodological integrity while also supporting accessibility and clarity.

Braun, V., Clarke, V., 2006. Using thematic analysis in psychology. Qual. Res. Psychol. 3, 77–101. https://doi.org/10.1191/1478088706qp063oa

6 – Include more details about the study’s limitations, such as selection bias or limitations of the qualitative method itself.

AU: Thank you for this valuable suggestion. In response, we have added a new subsection titled “Limitations and future research” in the discussion, where we expand on the limitations of our study and outline recommendations for future work.

We acknowledge the potential for selection bias, as participants were recruited through convenience and snowball sampling. Despite efforts to contact provincial producer organizations to broaden recruitment, we were unsuccessful in securing their support, which may have limited the diversity of participants.

We also recognize that conducting interviews by phone, while practical for reaching participants across Canada, limited opportunities for observing non-verbal cues and may have felt less personal. Additionally, while individual interviews allowed participants to speak freely, they do not offer the interactive benefits of group discussions where ideas can be exchanged and co-constructed. Future studies could complement interviews with focus groups to explore shared reasoning and social dynamics, or with surveys to enhance the representativeness and generalizability of findings.

These additions are now included in lines 682–712 of the revised manuscript.

7 – How can we be sure the participants gave their honest opinions and were not influenced by the way questions were asked? Is the interviewer experienced in interviewing?

AU: Thank you for this important observation. We acknowledge that the potential influence of the interviewer and the authenticity of participants' responses are inherent limitations of all qualitative research. To mitigate these concerns, the interviewer, who is experienced in conducting qualitative interviews and facilitating focus groups, took several steps to build rapport and create a comfortable environment that encouraged open and honest sharing. While portions of the rapport-building conversation were excluded from the transcript excerpts and supplementary materials to protect participant anonymity, they played an important role in fostering trust.

We have addressed this explicitly in the manuscript at line 187, which now reads: “All interviews were conducted by the first author (BV), an experienced interviewer who employed efforts to build rapport with participants before interviews began."

8 – Conclusion (lines 652–653) The first lines of the conclusion say the method was “effective,” but the study does not seem to evaluate the method itself. I suggest rephrasing this part.

AU: We agree that the original phrasing may have implied an evaluation of the method, which was not the study’s aim. We have revised the sentence to better reflect the intent and focus of the research. The revised sentence now reads (line 714): “This study employed semi-structured interviews to explore beef farmers' views on dairy-beef production, providing a deeper understanding of their perspectives on this practice.”

---

## [Decision Letter · Decision Letter 1]

5 Aug 2025

It’s not black and white: Perspectives of Western Canadian beef farmers on dairy-beef production.

PONE-D-25-22276R1

Dear Dr. von Keyserlingk,

We’re pleased to inform you that your manuscript has been judged scientifically suitable for publication and will be formally accepted for publication once it meets all outstanding technical requirements.

Kind regards,

Juan J Loor

Academic Editor

PLOS ONE

Additional Editor Comments (optional):

Reviewers' comments:

Reviewer's Responses to Questions

**Comments to the Author**

1. If the authors have adequately addressed your comments raised in a previous round of review and you feel that this manuscript is now acceptable for publication, you may indicate that here to bypass the “Comments to the Author” section, enter your conflict of interest statement in the “Confidential to Editor” section, and submit your "Accept" recommendation.

Reviewer #1: (No Response)

2. Is the manuscript technically sound, and do the data support the conclusions?

Reviewer #1: Yes

3. Has the statistical analysis been performed appropriately and rigorously? 

Reviewer #1: N/A

4. Have the authors made all data underlying the findings in their manuscript fully available?

Reviewer #1: Yes

5. Is the manuscript presented in an intelligible fashion and written in standard English?

Reviewer #1: Yes

6. Review Comments to the Author

Reviewer #1: (No Response)

7. PLOS authors have the option to publish the peer review history of their article (what does this mean? ). If published, this will include your full peer review and any attached files.

**Do you want your identity to be public for this peer review?** For information about this choice, including consent withdrawal, please see our Privacy Policy .

Reviewer #1: No

---

## [Editor Report · Acceptance letter]

PONE-D-25-22276R1

PLOS ONE

Dear Dr. von Keyserlingk,

I'm pleased to inform you that your manuscript has been deemed suitable for publication in PLOS ONE. Congratulations! Your manuscript is now being handed over to our production team.

Kind regards,

on behalf of

Dr. Juan J Loor

Academic Editor

PLOS ONE